# Ear Detection Using Convolutional Neural Network on Graphs with Filter Rotation

**DOI:** 10.3390/s19245510

**Published:** 2019-12-13

**Authors:** Arkadiusz Tomczyk, Piotr S. Szczepaniak

**Affiliations:** Institute of Information Technology, Lodz University of Technology, ul. Wolczanska 215, 90-924 Lodz, Poland

**Keywords:** geometric deep learning, ear detection, structured prediction, semantic segmentation, rotation equivariance, Gaussian mixture model, superpixels

## Abstract

Geometric deep learning (GDL) generalizes convolutional neural networks (CNNs) to non-Euclidean domains. In this work, a GDL technique, allowing the application of CNN on graphs, is examined. It defines convolutional filters with the use of the Gaussian mixture model (GMM). As those filters are defined in continuous space, they can be easily rotated without the need for some additional interpolation. This, in turn, allows constructing systems having rotation equivariance property. The characteristic of the proposed approach is illustrated with the problem of ear detection, which is of great importance in biometric systems enabling image based, discrete human identification. The analyzed graphs were constructed taking into account superpixels representing image content. This kind of representation has several advantages. On the one hand, it significantly reduces the amount of processed data, allowing building simpler and more effective models. On the other hand, it seems to be closer to the conscious process of human image understanding as it does not operate on millions of pixels. The contributions of the paper lie both in GDL application area extension (semantic segmentation of the images) and in the novel concept of trained filter transformations. We show that even significantly reduced information about image content and a relatively simple, in comparison with classic CNN, model (smaller number of parameters and significantly faster processing) allows obtaining detection results on the quality level similar to those reported in the literature on the UBEAR dataset. Moreover, we show experimentally that the proposed approach possesses in fact the rotation equivariance property allowing detecting rotated structures without the need for labor consuming training on all rotated and non-rotated images.

## 1. Introduction

A biometric can be defined as a measurable, physical characteristic, which can be used to identify individuals. There are various types of biometrics used in practical applications: voice recordings, fingerprints, signatures, DNA, hand geometry, iris and face images, or even keystroke dynamics, to mention only a few. A good biometric should have several properties [1]. It should be universal (everyone should possess this characteristic), distinctive (it should allow discriminating between people) and permanent (ideally, it should not change in time). Moreover, the process of acquisition should be inexpensive, generally acceptable, and not troublesome (in some applications, it should be even discreet). Finally, the identification system using such a biometric should be hard to circumvent. The biometrics mentioned above meet those expectations to varying degrees. It is relatively easy to forge a signature, whereas a DNA test usually is hard to falsify. Similarly, collecting face images, as a rule, is treated as a violation of privacy, while taking fingerprints seems to be natural.

In this work, ear images are considered [2,3]. There are several factors that make the research and application of ear recognition important and attractive: people can be identified on that basis; this biometric does not change in time; and its gathering does not create a great deal of controversy. Furthermore, technology enables acquisition of ear images from a distance, which may be of great importance for police investigations and in security systems. It has, however, at least two consequences. Firstly, before the individual can be identified [4,5], the localization of the ear must be precisely detected. Secondly, since the acquisition process is not controlled, the orientation of the head can significantly vary. As a result, the need for detection methods arises, which will be able to cope with ear transformations.

Convolutional neural networks (CNNs) became a state-of-the-art solution of many image analysis problems [6,7]. Their main component is convolutional layers designed to apply the same, trainable filters (represented by a rectangular mask) locally to every part of the image. It allows extracting the spatial distribution of image characteristic features (so-called feature maps). These kinds of layers together with downsampling/upsampling mechanisms and classic fully connected layers allow solving many typical tasks. These are, among others: image classification, object localization and detection, semantic and instance segmentation, etc. However, despite unquestionable advantages, convolutional neural networks are not free of vices. First of all, they operate on pixels. Bearing in mind the dimensions of currently processed images, to solve practical problems, the structure of such networks must be quite complex (deep architectures). This results in a large number of parameters that need to be trained and, consequently, huge training datasets that need to be prepared. The second problem is their sensitivity to object rotations. If rotation invariance/equivariance is required, the complexity of the trained model must be further increased. Both of those problems can be overcome by the geometric deep learning (GDL) technique presented in this paper.

### 1.1. Ear Detection

In [8], the authors emphasized the extremely challenging character of ear detection as this part of the human head can be presented on images in various sizes, rotations, shapes, and colors. Moreover, the images can be of diverse quality, and fragments of ears can be partly hidden. To overcome and manage those problems, and to offer solutions that are applicable in practice, in the last few years, machine learning methods have become more and more popular in use. Motivated by the current direction of research development, in the following, we continue the surveys [2,3] and present a short, but comprehensive view on progress made since 2016 in machine learning application to the problem under consideration.

The approach published in [9] was based on geometric morphometrics and deep learning. It was proposed for automatic ear detection and feature extraction in the form of landmarks. A convolutional neural network (CNN) was trained and results compared with a set of manually landmarked examples.

A two step approach was described in [10]. In the first step, the detection of three regions of different scales allowed obtaining information about the ear location context within the image. In the second step, a filtering was performed with the aim of extracting the correct ear region and eliminating the false positives ones. This technique used convolutional neural networks (called here multiple scale Faster R-CNN) to detect ears from profile images.

In [8], the authors applied a convolutional encoder-decoder network to perform the binary classification of image pixels as belonging to either the ear or to the non-ear class. Temphhe result was improved by a post-processing procedure based on anthropometric knowledge and deletion of spurious image regions. The paper involved comparative results with a state-of-the-art known from the literature.

A detection technique applying an ensemble of convolutional neural networks (CNNs) was presented in [11]. The weighted average of the outputs of three trained CNNs was considered as result of detection of the ear regions. A better performance observed for the ensemble of networks compared to the use of single CNN models was reported. A similar approach was described in [12], where also an ensemble of three networks was used. This time, however, members of the ensemble did not differ in network architecture, but they were trained with regions of the image taken at different cropping scales.

### 1.2. Rotation Equivariance

Transformation invariance and equivariance are terms sometimes mistakenly used interchangeably. The first means that the system will respond in the same way regardless of the transformation, which is applied at its input, while the latter indicates that transformation of the input will result in the same transformation of the output. In the context of image analysis, if the classification task is considered, transformation invariance can be expected. In other problems like semantic segmentation discussed in this work, the desired property of the system is its transformation equivariance.

CNNs manifest natural translation equivariance since the same filters (feature detectors) are applied at different locations of the image. Thanks to the additional pooling operations, they also possess approximate invariance to translation. Input rotation, however, is still a problem for those networks.

Three main groups of techniques used to overcome it can be distinguished. The first one uses data augmentation, generating rotated versions of the training images, forcing the network to learn all the possible orientations of the objects. That approach, of course, requires very complex network architectures having sufficient flexibility. Two alternative methods use either an input image [13,14] or filter [15,16] rotations with some kind of result aggregation. In addition, to avoid excessive increase of the parameter number, trained weights are shared between different processing paths. The rotation of images and filters in classic CNNs in general is problematic as both are represented by a regular, rectangular grid of values. Consequently, some interpolation algorithms must be used, and the object of interest should be located in the image center to avoid artifacts at the border.

In [13], the authors combined augmentation with input image rotation. In the latter case, only $0^{\circ }$ and $45^{\circ }$ angles were considered together with additional image cropping and flipping. As a result, 16 variants of the same image were processed by the network. The output feature maps were concatenated and passed to dense layers serving as a classifier. A slightly different approach was presented in [14]. Here, also input image rotations were used, but this time as an aggregation method, transformation invariant pooling was used where the element-wise maximum was taken from resulting feature maps.

The authors of [15] rotated filters instead of input images. In this case, bicubic interpolation was applied to generate a group of rotated filters. As a response for each group, max-pooling was used. While training, gradients were passed through the element in a filter group with the largest activation. Such an orientation pooling could also be found in [16]. This time, however, to avoid interpolation, a set of dedicated atomic, circular filters were prepared, and the actual network filters were sought for as a linear combination of these.

At the end of this short survey, one more approach should be mentioned, as it does not fit any of the above-mentioned categories. In [17], the authors decided to embed inside CNN additional processing blocks, transforming, in particular rotating, feature maps. After transformation, the outputs were concatenated and processed by the further part of the network.

### 1.3. Geometric Deep Learning

GDL is a dynamically developing area of research in recent years [18]. It, inter alia, tries to generalize and apply the concept of CNN for structures less regular than images (graphs) and for continuous domains (manifolds). This adaptation requires the proper definition of the convolution operation, which should be able to compute features of given elements based on their local neighborhood. GDL was successfully applied to various practical problems. Two of the most popular fields of application are: prediction of chemical molecules’ properties [19,20] and document classification taking into account citation links [21,22]. In the first case, final prediction is assigned to a graph as a whole, while in the second, every graph node is considered separately. Surprisingly, very few of the approaches were tested on images. In most of the cases, the problem was the initial definition of graph convolution itself, which required a fixed structure of the graph (in the case of images, graphs are different depending on the content). The existing approaches were used only for handwritten digit classification (MNIST dataset) [23,24,25], where either a grid of pixels was treated as a graph or image content was represented by an irregular graph of superpixels [26].

The latter approach, although not popular in the GDL community, has undeniable advantages. The change of image representation, where its content is described with a significantly smaller set of spatially distributed elements, leads to a reduction of model complexity, which is required for its processing. Moreover, such a representation is more human friendly. Conscious understanding of image content operates rather on regions and borders separating them than on thousands of millions of pixels. This, in turn, enables simpler interpretability of the results and simpler acquisition of additional expert knowledge there, where the number of training samples is limited (e.g., medicine, biometry, etc.).

### 1.4. Contribution

The main contribution of this paper lies in the application of GDL to semantic segmentation of the images and in the introduction of trained filters’ rotations. Both of those features are illustrated with the ear detection problem, but can also be applied in other object detection, semantic segmentation, or image classification tasks.

The proposed approach to object detection is a novel area of application of GDL in image the analysis domain since, so far, in [24] and in similar works, these kinds of networks were used only to assign labels to image as a whole. When graph nodes were interpreted separately, as was done in this work (Figure 1), only the other application areas were explored in [21,22]. What is more, the originality of the method lies also in a specific approach to semantic segmentation itself. Typical applications of classic CNNs in this field use some downsampling/upsampling layers to reduce the number of parameters [27,28,29]. Only in a few specific applications described in [30,31], such mechanisms were not required. In this work, such techniques are not used as well, because reduced image representation allows using relatively simple models.

The proposed method is, of course, new in the context of ear detection as well. Until now, CNNs were applied only at the pixel level both for semantic segmentation [10] and direct object detection [8]. Here, alternative superpixel representation is used, showing its usefulness in these kinds of applications and allowing significantly simplifying the architecture of the trained network.

The additional novelty of this work lies in proving that training of the proposed model with a limited number of samples and rotation of trained filters allowed detecting the rotated structures as well (Figure 2). Consequently, after a simple training process, we could obtain the rotation equivariance property of the considered network. This approach is different from the filter rotations described in [15] or [16], as there these filters are also rotated when the network is trained, which complicates the whole procedure. It should be also emphasized that, since filters are defined by GMM, there are no interpolation problems, typical in classic CNNs, when filters are rotated.

The above described property can be useful in ear detection problems where profile images are acquired. If a limited number of training samples, in particular in only one orientation, can be gathered, we can prepare a rotation equivariant detection model, which should able to locate ears when the head is rotated in the image plane.

The content of this work is split into several sections. In Section 1, a short literature review in the areas of ear detection and rotation invariance/equivariance is presented. It contains also a description of the paper’s novelty and contribution. Section 2 contains the details of the proposed approach, as well as the results of an experiment verifying its properties. In Section 3, results and a detailed discussion of the main experiments are described. A summary of the conducted research concludes the paper.

## 2. Materials and Method

This section presents the characteristics of the dataset used, as well as the GDL technique discussed in this work. Naturally, to apply a convolutional neural network on graphs, additional pre- and post-processing had to be performed. The whole processing flow is shown in Figure 1; however, the majority of the details of the methods used will be presented in sections devoted to the specific experiments. Here, only the functioning of the neural network will be explained. Moreover, while describing the details of the GDL approach, the hypothesis about its properties will be formulated. That is why, at the end of this section, the results of the experiment, verifying the correctness of the proposed assumptions, are described as well.

### 2.1. Dataset

There are several publicly available benchmark datasets dedicated to ear biometric tasks. They differ, however, significantly. The CP dataset [33] and the IITD dataset [34,35] are very similar. Both contain grayscale, tightly cropped, and aligned images. The AMI dataset [36] and the WPUT dataset [37] are comprised of color images with ears and surrounding head fragments. In all those sets, images were acquired in controlled, laboratory conditions and can be used for training purposes and evaluation of ear identification systems. The last two sets described below were prepared in a different way. The UBEAR dataset [32] is composed of profile, grayscale images taken from video sequences (selected frames). Images in the AWE dataset [38] were collected from the web. They are better suited for ear detection problems as they contain also original, not cropped, pictures. It should be also emphasized that, except the first two, in all those datasets, to a different extent, some additional difficulties are present. To mention only a few of them: images were acquired in different illumination conditions and with different background; ears were occluded; head was rotated, leading to ear transformations, etc.

The UBEAR and AWE datasets together with sets dedicated to other problems (e.g., face recognition) are exploited in the ear detection literature. In this work, in all the experiments, only the UBEAR dataset was used. There are several reasons for that choice. First of all, we did not have access to the original, not cropped, AWE images. Secondly, it corresponded to our initial idea of an ear detection system where people could be identified discreetly from video sequences. Thirdly, this set is relatively large. It contains 4429 images taken from 126 persons. Fourthly, in this dataset, binary masks, indicating precise ear positions, are available, which is a rare case in these kinds of datasets (usually only bounding boxes are annotated). What is more, it contains information about head poses, which allows identifying images with a specific head orientation. There were 5 poses identified. Every pose had a unique letter assigned: M means that person was stepping ahead (normal head orientation), whereas U, D, O, and T indicate that the head was rotated upwards, downwards, outwards, and towards, respectively. Finally, it is a quite challenging dataset for analysis. Not only all the above mentioned difficulties were present, but also motion artifacts could be observed. Sample images from the UBEAR dataset are presented in Figure 3.

### 2.2. Method

In this work, the existing GDL method, presented in [24], was further developed. That approach defined convolutional filters using Gaussian mixture model (GMM) in a pseudo-coordinate space. Assuming that nodes of input and output graphs are described with vectors in *N* and *M* dimensions (channels), respectively, the operation of a single convolutional layer ϕ can be expressed in the following way (Figure 4):(1)$$h^{m}\left(s\right)=\Psi \left(b^{m}+\sum_{n=1}^{N}\sum_{t\in N(s)}\varphi^{n,m}\left(u\left(s,t\right)\right)f^{n}\left(t\right)\right)$$
where:(2)$$\varphi^{n,m}\left(u\right)=\sum_{j=1}^{J}g_{j}^{n,m}\exp\left(-\frac{1}{2}{\left(u-\mu_{j}^{n,m}\right)}^{T}{\left(K_{j}^{n,m}\right)}^{-1}\left(u-\mu_{j}^{n,m}\right)\right)$$
and m=1,…,M. In the above equations, *s* and *t* are node indices, *f* and *h* represent vectors describing features of the input and output graphs’ nodes, respectively, *J* denotes the number of Gaussians, and N is the neighborhood function identifying adjacent nodes. Mapping u calculates pseudo-coordinates of node *t* relative to given node *s*. These coordinates are *d*-dimensional vectors. Finally, Ψ is an activation function applied element-wise for every graph node, and *b* represents additional, optional bias. Every convolutional layer defined in this way contains *M* groups of *N* filters φ. The trainable parameters of those filters are: real numbers *g*, vectors μ of size *d*, and diagonal d×d matrices K (only *d* non-zero elements). This gives a total number of parameters per filter equal to J(2d+1) and in the whole layer equal to MN(J(2d+1)+1).

The above formulation differs slightly from the original one presented in [24]. In that paper it was not clear whether every pair of input and output channels had its own fully trainable filter. In our experiments we have used PyTorch Geometric library [39] where, in its earlier versions, some of the filter parameters were shared. The presented extension was added to the library by the authors of this work and is available in PyTorch Geometric starting from version 1.3.1.

To construct a network operating on graphs and useful for semantic segmentation tasks, we must ensure that input and output graphs have the same size and structure. In classic CNNs, to achieve that goal without excessive growth of the number of parameters, additional downsampling/upsampling blocks are used. Here, thanks to the reduced representation of the image content, it was sufficient to consider only a sequential composition of the above layers:(3)$$h=\Phi \left(f\right)=\left(\phi^{L}\circ \ldots \circ \phi^{1}\right)\left(f\right)$$
where *L* denotes the number of layers. Naturally, the number of input *N* and output dimensions *M* (as long as they are consistent in successive layers), as well as activation functions Ψ may vary between layers.

Every filter φ is described by the corresponding GMM in pseudo-coordinate space defined by mapping u. When Cartesian coordinates are used in the image plane (d=2), GMM can be rotated around the origin of the coordinate system (0,0) by an angle θ, resulting in a new filter $\varphi_{\theta }$. If the original filter φ detects some specific node configurations in a graph, the filter $\varphi_{\theta }$ should have a high response for those nodes, which exhibit the same, but rotated characteristic of their neighborhood. Consequently, if all the filters in convolutional layers $\phi_{\theta }$ are rotated, the whole network $\Phi_{\theta }$ should possess the same property. To confirm this hypothesis, a verification experiment, described in the next subsection, was proposed. This confirmation is crucial to facilitate the construction of the system possessing the rotation equivariance property shown in Figure 2. We can train then a network $\Phi_{0}$ capable of recognizing ears for basic head pose (a smaller training set is required) and, after successive rotations of filters, use it to detect rotated ears as well.

### 2.3. Verification

To verify the hypothesis about the capability of the trained model to detect rotated structures, the experiment presented below was conducted.

First, from UBEAR dataset [32], fragments of binary masks with precise ear localization (100 samples) were extracted. Next, their graph representations were created using the SLIC algorithm [26]. Nodes of those graphs corresponded to generated superpixels and were described by their average intensity normalized to the [0,1] interval (Figure 5a). Directed edges connected superpixels’ centroids (Figure 5b) and had Cartesian pseudo-coordinates assigned.

Those graphs allowed generating a family of training $D_{\theta }^{TR}$, validation $D_{\theta }^{VA}$ and test $D_{\theta }^{TE}$ sets with 60, 20, and 20 samples, respectively, where for every input graph, the expected output graph was prepared. The values assigned to nodes of output graphs indicated whether the node approximately corresponded to an outer edge at θ angle. They were calculated using the following formula:(4)$$h_{\theta }\left(s\right)=\max\left(\frac{1}{\Omega_{\theta }\left(s\right)}\sum_{t\in N\left(s\right):\omega_{\theta }\left(s,t\right)>0}\omega_{\theta }^{2}\left(s,t\right)\left(f\left(t\right)-f\left(s\right)\right),0\right)$$
where:(5)$$\Omega_{\theta }\left(s\right)=\sum_{t\in N\left(s\right):\omega_{\theta }\left(s,t\right)>0}\omega_{\theta }^{2}\left(s,t\right)$$
and:(6)$$\omega_{\theta }\left(s\right)=\sin\left(\alpha \left(s,t\right)-\theta \right)$$

As before, *f* and *h* represent feature vectors of input and output graphs’ nodes and α(s,t) is the angle between the horizontal axis and edge connecting nodes *s* and *t*. The sample, horizontal edge found in this way for θ=0 is depicted in Figure 5c.

Having these data prepared, the CNN with GMM filters was trained using only $D_{0}^{TR}$. The MSE loss and Adam optimizer with the learning rate equal to $10^{-3}$ were applied. The validating samples were used to check if the model was not overfitted and to select the optimal one. The network contained L=2 convolutional layers. There were 10 groups with 1 filter in the first layer $\phi^{1}$ and 1 group with 10 filters in the second one $\phi^{2}$. Every filter contained J=4 Gaussian functions. In the first layer, ReLU and, in the second, identity activation functions were used. In Figure 6, the sample trained GMM filter φ from the first layer together with its rotated version $\varphi_{\theta }$ are presented. The results depicted in Figure 7 prove that the proposed concepts behaved correctly not only for training samples, but they were also able to generalize and give reasonable results for unseen graphs. In Table 1, a systematic evaluation is presented. It is evident that if filters in the original network $\Phi_{0}$ were rotated by an angle θ, the resulting network $\Phi_{\theta }$ was able to detect structures rotated by the same angle (significantly smaller MSE error) effectively.

## 3. Results and Evaluation

This section contains the results of the experiments conducted on the UBEAR dataset [32]. First, the convolutional network was trained to detect ears in images with a normal head orientation. Next, it is applied for the detection of ears for other, selected head poses. At the end of this section, the discussion of the experiments’ outcomes is presented.

### 3.1. Assumptions

To apply the proposed approach, the content of every image in the UBEAR dataset needed to be represented as a graph. For that purpose, first, images and binary masks with precise ear localization were scaled down to have only 0.25 of their original size. Next, superpixel detection was performed with the use of the SLIC algorithm [26]. The expected number of superpixels, which is a parameter of SLIC algorithm, was determined by their expected average area *A*. Two possible configurations were considered with A=256 and A=128. They led to around 300 and 600 superpixels per image respectively.

Having superpixels generated, two graphs were created: input graph and expected, output graph. Nodes of both graphs corresponded to superpixels. In the input graph, the feature vector assigned to node contained the average intensity of image pixels covered by a given superpixel (Figure 3a,b). In the expected, output graph, it was the average intensity of pixels taken from binary masks multiplied by scaling constant W>0. It should be noted that in the latter case, values assigned to nodes need not be equal either to zero or *W* as the borders of superpixels need not coincide with the borders of the ear region (Figure 3c,d). Two possible values of scaling constant were considered. These were W=1 and W=100. This constant was introduced based on our earlier experience with classic CNN applications. Such a procedure allowed avoiding, during network training, local minima with all responses equal to zero.

Nodes in the considered graphs must be connected with directed edges. To determine which airs of nodes should be connected, first, the adjacency of all superpixels was examined. Two nodes were connected with an edge if there existed a path (path in this context means a sequence of superpixels) of length shorter than or equal to a given number *D*, connecting corresponding superpixels. Here, also two configurations were analyzed, where D=1 and D=2 (Figure 8). Selecting a higher value of *D*, we should be able to increase the size of the visual field, i.e., increase the number of input nodes, which influences the single output node. Loops connecting nodes were allowed because thanks to that the node could express its influence on the output assigned to this node.

After a series of initial trials, a network architecture with L=4 layers ϕ was found to be the optimal one. The number of filter groups in those layers, and hence the number of output graphs, was equal to 20, 10, 5, and 1, respectively. The number of filters in the given group corresponded to the number of layer inputs. In the first layer, it was one, and in the subsequent layers, it depended on the output of the previous layer. In all layers except the last one, the ReLU activation function was used. The last layer had an identity activation function assigned. The number of GMM components in all filters φ was equal to J=4. As before, while training, MSE loss, as well as the Adam optimizer were used. This time, however, a smaller learning rate, equal to $10^{-4}$, was considered. For weight initialization, the Glorot scheme was used [40].

To detect ears based on the output of the network, simply the node (superpixel) with the highest response was sought (Figure 9a,b). In order to evaluate if this detection was correct, it was checked whether the superpixel (its centroid) lied inside the bounding box surrounding the ear region. That rectangle was found using the original binary masks provided in the UBEAR dataset and was slightly enlarged to take into account the size of the superpixels (Figure 9c,d).

In all the experiments, images from the UBEAR dataset were split into three sets: training $D^{TR}$, validation $D^{VA}$, and test set $D^{TE}$. The split was made made based on person identifiers, i.e., images of the same person were assigned always to the same set. This should allow checking if the trained models were able to generalize the acquired knowledge and respond correctly for new people. The validation set was used to prevent overfitting by the selection of the optimal model from among models created in the training phase. The number of people in the discussed sets was equal to 75, 25, and 26, respectively. Only left ears were considered since the proposed approach should work for right ears in the same way. What is more, mirror transformation of GMM filters should also allow applying the network trained with left ears to work for right ears, as well.

### 3.2. Experiment I

In the first experiment, the CNN network was trained on a UBEAR subset $D_{M}^{TR}$ containing only heads in their standard orientation (pose M in the UBEAR dataset). Every combination of parameters *A*, *D*, and *W* was tested to select the optimal one. The obtained results and cardinalities of the considered training, validation, and test sets are gathered in Table 2. In all cases, results were satisfactory with the correct detection rate bigger than 90%. It is worth noting that in three cases, the detection accuracy for the training set was equal to 100%. This seemed, however, to be slightly overfitted, and as the best configuration, the one with A=256, D=2, and W=100 was indicated.

The closer analysis of the results revealed that, surprisingly, representation with A=256 was not worse than representation with a bigger number of superpixels obtained when A=128. It could be expected that in the latter case, when more details are given, the accuracy would increase. The explanation can be the fact that in both cases, the same network architecture was used, and consequently, for smaller *A*, the effective visual field was also smaller. What is more, in both cases, the same number 2000 of training iterations was used, and perhaps, more details required longer training. Nevertheless, since for A=256, the results were satisfactory and, thanks to the simpler representation, graph processing was faster, this seemed to be a reasonable choice for the discussed problem. In the case of other parameters, the configurations with D=2 and W=100 seemed to lead to models with better generalization abilities. For them, the detection accuracy was higher when validation and test sets were considered. Those observations were also confirmed by the training characteristic depicted in Figure 10. For optimal values of the parameters (Figure 10a), the best model, the one with the smallest validation error, could be easily selected in the early stage of training. In other cases (Figure 10b), both errors seemed to decrease slowly, and maybe, further training could provide a better solution. Not without significance is also the random initialization of network weights.

In Figure 11, additional samples of good and wrong detections of the optimal network are presented. It can be observed that correct detections were possible in different illumination conditions and with different background, as well as in situations when there were additional objects located in the ear neighborhood. The typical reasons for the detection mistakes were: questionable annotations (head orientation slightly different than expected), position of the ear close at the image border, and highly occluded ears.

To have a better insight into the process of single graph processing, also the selected outputs of convolutional layers ϕ are presented in Figure 12. In classic CNNs the first layers are usually responsible for the detection of some local image characteristics. Although here, the interpretation was not that obvious, that kind of behavior to a certain extent can also be observed. In Figure 12a, for example, the detection of vertical edges seemed to take place.

### 3.3. Experiment II

In the second experiment, the best network, trained to detect ears in their basic orientation (pose M), was applied to detect ears when head was rotated in the image plane (poses U and D). This network, trained only using samples from $D_{M}^{TR}$, will be further denoted as $\Phi_{0}$ to indicate that it was not rotated. In Table 2, initial detection results for subsets of $D_{U,D}$ obtained with this network are presented. These results, between 80% and 90%, were surprisingly good. Three explanations seem to be possible. Firstly, apparently, in the UBEAR dataset, most of the cases with U and D pose were similar to pose M (the head rotation was not large). Secondly, ear orientation relative to the head is an individual feature. Slightly rotated ears in $D_{M}^{TR}$ could allow $\Phi_{0}$ to learn how to recognize them in $D_{U,D}$. Finally, which in general is an interesting hypothesis, to detect ears, it may be sufficient to observe only the configuration of image regions containing head. Humans need not see ear details to make correct detections. Perhaps, our CNN, working on reduced graph representations, did the same thing.

To check if network rotations can help in the detection of rotated ears, we prepared a set containing 11 networks $\Phi_{\theta }$. They were created based on network $\Phi_{0}$ where filters φ were rotated by angles θ equally distributed in interval [−π,π]. Of course, the original network $\Phi_{0}$ was in this set. Next, every image was processed by all those networks, and that output was considered to be the final result, for which the maximum node value was observed (Figure 13). It was expected that such system would possess the rotation equivariance property, i.e., the network $\Phi_{\theta }$ with correct angle θ correlated with ear orientation should give the rotated response of the network $\Phi_{0}$ for basic ear orientation.

The results obtained in this way are presented in Table 3. To our surprise, they were worse than the result of separate network $\Phi_{0}$. This means that the network rotations introduce additional maxima in the wrong regions of the image. They can be observed in Figure 13 and Figure 14. Two typical reasons for incorrect detections are presented there. Firstly, areas, which locally, at a certain angle, can be considered similar to the ear, were indicated (Figure 14f). This behavior could be expected and cannot be avoided at this level of image representation. Secondly, there are maxima in completely unexpected locations (Figure 14b). We have a suspicion that those artifacts are caused by the characteristic of superpixels generated by the SLIC algorithm in regions of uniform color. They were very regular and, since the network $\Phi_{0}$ was trained only for one orientation, it was not able to give correct responses in situations when the graph was rotated.

After further analysis of the results, we also noticed that even if the output with maximum node value did not allow improving the detection results, the ear localization was frequently indicated correctly by one of the networks $\Phi_{\theta }$ (Figure 14). To check if this was a general rule, we conducted an additional experiment where the results were accepted (correct detection) if any network $\Phi_{\theta }$ was able to solve the task. Those results are also shown in Table 3. Accuracy calculated in this way, for all poses, was above 96%. It proved that the required information was not lost and rotations of the trained filters allowed constructing a satisfactory solution.

### 3.4. Discussion

In the literature, several results can be found concerning the UBEAR dataset. In [10], authors reported the accuracy of their method to be equal to 98.22%. This is a significantly better result than other, mentioned in that paper, techniques. The traditional Faster R-CNN was able to reach 65.56%, whereas AdaBoost gained 51.74%. An ensemble of classic CNNs, described in [12], achieved an accuracy equal to 75.08%. Our results cannot be directly compared with these values because of several reasons. Firstly, in the mentioned papers, models were trained using data coming from datasets other than UBEAR. Secondly, other methodologies were used to indicate detections (bounding boxes), and consequently, other evaluation measures had to be used to verify their correctness. Thirdly, we took into account only left ears assuming that for the right ones, our approach would behave in a similar way. Finally, our network was trained using only one pose of the head.

That is why, to show convincingly and objectively the quality of our method, we decided to train our model using set $D_{ALL}^{TR}$ with all the poses. The obtained results are included in Table 3. The network $\Phi_{ALL}$, trained in this way, was able to recognize images in the training, validation, and test sets with accuracies of 98.75%, 92.91%, and 94.15%, respectively, which were very satisfactory. Nevertheless, since the training data were more complex, to further improve them, we extended our basic architecture by an additional layer on the input of the network. This layer consisted of 20 filter groups. The results of that network, denoted as $\Phi_{ALL}^{+}$, are also shown in Table 3. One can observe an improvement, in particular for images in validation set $D_{ALL}^{VA}$.

The presented results revealed also indirectly that there was redundant information in image pixels. In order to detect ears, it was not required to operate on millions of pixels, which usually leads to very complex models. Classic CNNs, usually used for semantic segmentation (e.g., FCN [27], DeepLab [28], or SegNet [29]), require a big computational effort because they have tens of millions trainable parameters. Our network had only several thousands of parameters. This, in turn, sped up both the training and processing of single images. On the same CPU, the forward pass of one graph took on average 0.14 s, while FCN processed the one scaled-down image about 15.6 times slower, i.e., in 2.19 s. Even taking into account the time required to transform the image into a graph (superpixel generation with the use of SLIC algorithm lasted 1.65 s), our approach allowed getting very good results significantly faster.

## 4. Summary and Future Work

In this work, it was proven that the application of CNN operating on graphs for semantic segmentation allowed effectively solving a biometric task of ear detection. The best trained model was able to achieve more than 94% (depending on the considered subset) accuracy on the UBEAR dataset, which contained images with different head orientations, different illumination conditions, and where occlusions and motion artifacts were present as well. This result (Table 3) was comparable with the best results reported in the literature on the same dataset. What is more, the reduced, superpixel based image representation (hundreds of superpixels instead of millions of pixels) allowed constructing a relatively simple model (fewer parameters), which processed data faster than classic CNNs.

We also showed that the specific, used in this paper, network with GMM filters could be used to construct a system having the rotation equivariance property (Figure 2). It can be trained with a limited amount of data, where only structures in one orientation are available and no augmentation is used. The experiments revealed that such a model potentially allows achieving results even better than the model trained using structures in all their possible orientations (Table 3). Additional investigation is, however, required to understand and filter out the artifacts that appear.

The superpixel based image representation used in this work is not the only option. We are currently exploring other alternatives focusing not only on the character of the elements describing image content, but also on their faster generation. This and the short processing time of our networks should allow constructing systems able to detect ears efficiently on devices with only a CPU available.

Another interesting research direction is further theoretical analysis of the presented approach. It was shown that even a very coarse representation (relatively small number of superpixels) of image content allowed getting satisfactory results. This is not fully surprising since humans can do that with 100% accuracy for images in the UBEAR dataset. Apparently, ear details are not required to detect them correctly. It is suspected that for humans, it is enough to identify only the region containing head. Further research should show if such a mechanism also takes place in our networks. All the more, such an analysis, thanks to the reduced image content representation, should be easier than the analysis of classic CNNs. Firstly, this is because, we were not operating on a huge set of pixels, and consequently, the analyzed network was simpler (only a few layers are enough to cover a large visual field). Secondly, this is because humans are not operating consciously directly on pixels, and the explanation of the algorithm behavior in terms of easily understandable, small, and homogeneous regions will be more natural and convincing. That potential of explainability can be considered as an additional advantage of the presented technique.

## Figures and Tables

**Figure 1 sensors-19-05510-f001:**
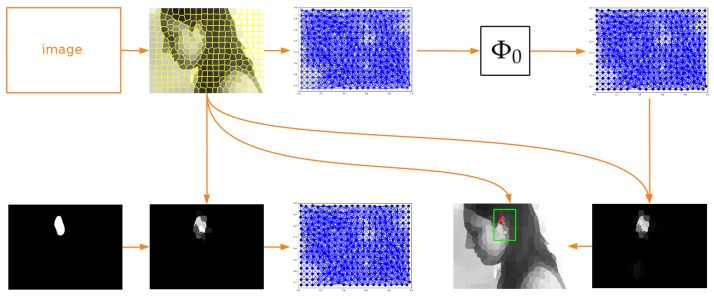
Image processing flow of a convolutional network operating on graphs. As an input, the image from the UBEAR dataset [32] (it is not shown here to protect the identity of the depicted person) and corresponding binary mask, indicating precise position of the ear, are given. In the first step, the SLIC algorithm [26] is used to transform image content into its superpixel representation. Next, taking this representation and original mask into account, the expected values are assigned to superpixels. They indicate the correspondence between superpixels and the ear region and constitute the goal of training (second column). Further, graphs for both the transformed image and expected mask are constructed (third column). The edges connecting nodes are created based on the spatial adjacency of superpixels. It is worth noticing that the expected values are assigned to graph nodes and not to the graph as a whole. Those graphs are used to train network $\Phi_{0}$, which learns to localize ears for basic head orientation. When the network is trained, it can process any other input graph. Knowing the output graph and the distribution of superpixels, it is possible to construct the output mask where the maximum should reflect the position of the ear (last column).

**Figure 2 sensors-19-05510-f002:**
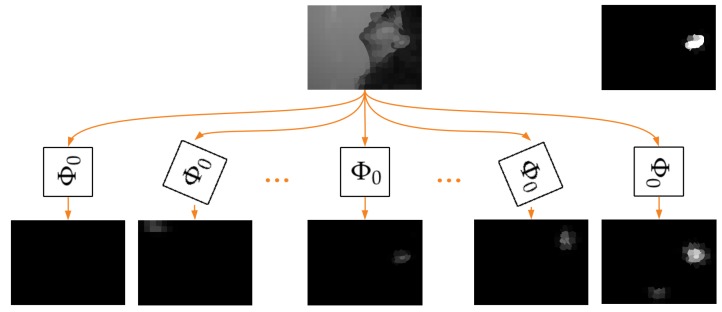
System with the rotation equivariance property. Network $\Phi_{0}$ is trained using only samples with basic head orientation (Figure 1). After rotation of GMM filters, it is able to detect the corresponding rotated structures as well. Consequently, selecting the output mask with the maximum value as the output of the whole system, we can obtain the desired rotation equivariance property. Thanks to that, we are able to localize ears even for non-standard head orientations. The top right image presents the expected output mask.

**Figure 3 sensors-19-05510-f003:**
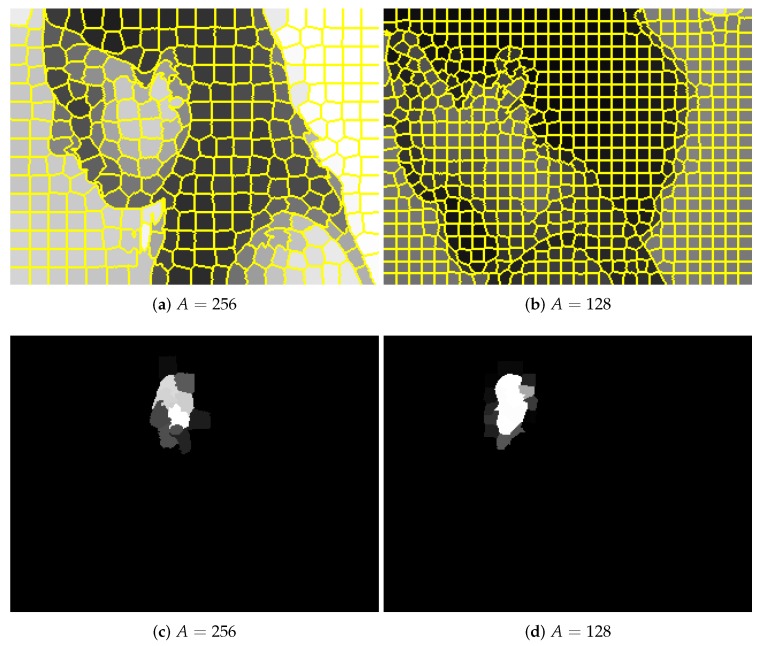
Sample images from the UBEAR dataset (pose M). Their content was described using superpixels with different average areas *A* of a single superpixel (here and in the whole work, the original images are not used on purpose to protect the identity of the depicted people): (**a**) original image, (**b**) binary mask with ear localization. The color assigned to every superpixel is the average color of covered pixels.

**Figure 4 sensors-19-05510-f004:**
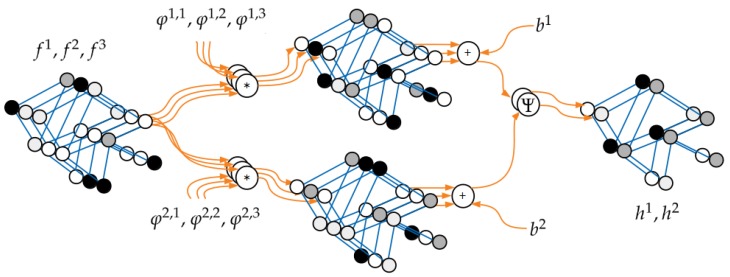
The processing scheme of a single layer ϕ. Here, the input and output graphs are described with vectors in N=3 and M=2 dimensions, respectively. For illustration purposes, every dimension (channel) is shown separately, but of course, the graph structure is always the same. It can be observed that every pair of input $f^{n}$ and output $h^{m}$ channels has its own fully trainable GMM filter $\varphi^{n,m}$.

**Figure 5 sensors-19-05510-f005:**
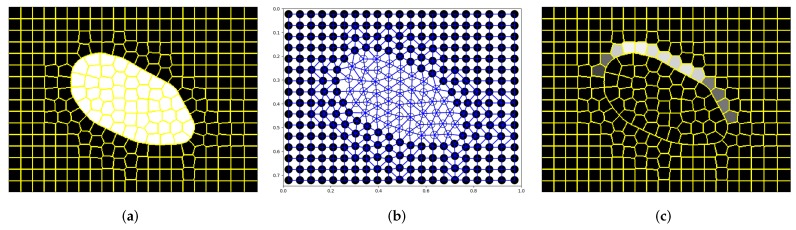
Images and graph used for initial verification of the proposed approach: (**a**) original image with superpixels detected; (**b**) its graph representation; (**c**) the expected output. In this experiment, the expected average size of the superpixel was equal to A=400, and graph nodes were connected only if the corresponding superpixels were adjacent. It is worth noticing that the irregularities of superpixels and consequently irregularities of the graph structure are present only if image colors are not uniform. This is a typical situation when the SLIC algorithm is used for superpixel generation.

**Figure 6 sensors-19-05510-f006:**
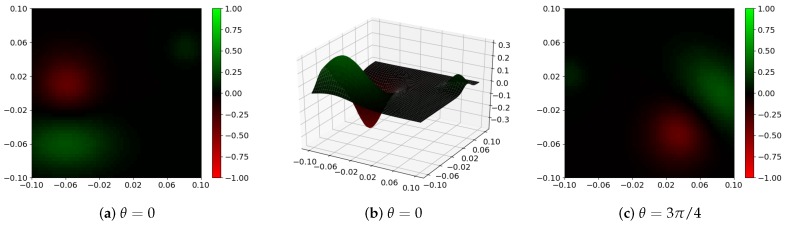
Sample GMM filter in layer $\phi^{1}$ of the CNN trained in the described experiment: (**a**,**b**) 2D and 3D filter visualization, respectively; (**c**) GMM filter rotated by an angle θ. In all cases, red color represents a negative value, and green color represents a positive one. Black color corresponds to values close to 0.

**Figure 7 sensors-19-05510-f007:**
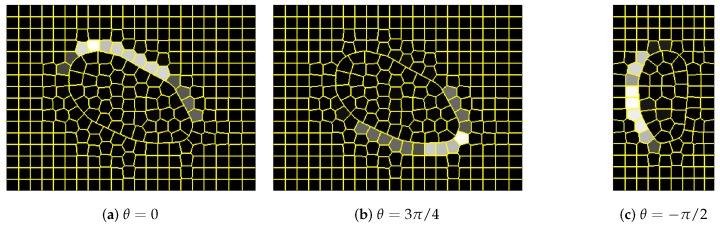
Outputs of the trained CNN for different filter rotation angles θ: (**a**,**b**) results for training image shown in Figure 5; (**c**) result for the image from test set. It should be noticed that network $\Phi_{0}$ trained to detect structures in their basic orientation is able to give a reasonable answer when its rotated version $\Phi_{\theta }$ is used.

**Figure 8 sensors-19-05510-f008:**
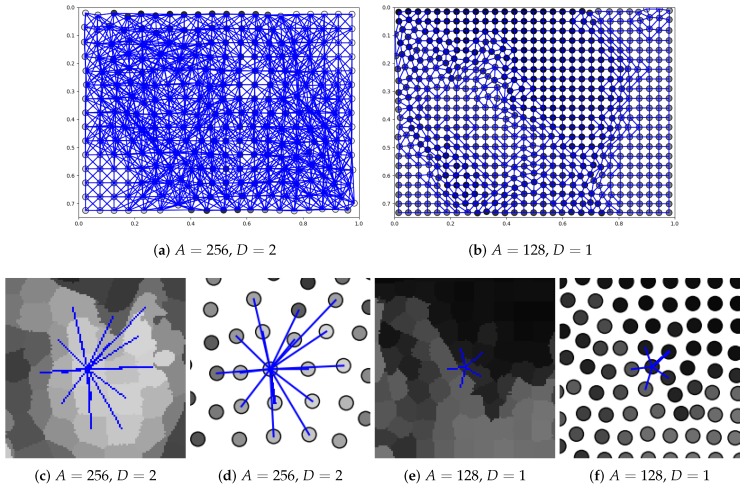
Graphs generated for images shown in Figure 3 for different superpixels’ number and different node neighborhoods (parameters *A* and *D*, respectively): (**a**,**b**) full graph with all edges; (**c**,**e**) superpixels with the local neighborhood of the selected node; (**d**,**f**) selected graph node with its neighborhood. As the image scale was preserved, it can be observed that when *A* was smaller, the smaller image region was processed by CNN. Consequently, not only *D* but also *A* influenced the size of the effective visual field.

**Figure 9 sensors-19-05510-f009:**
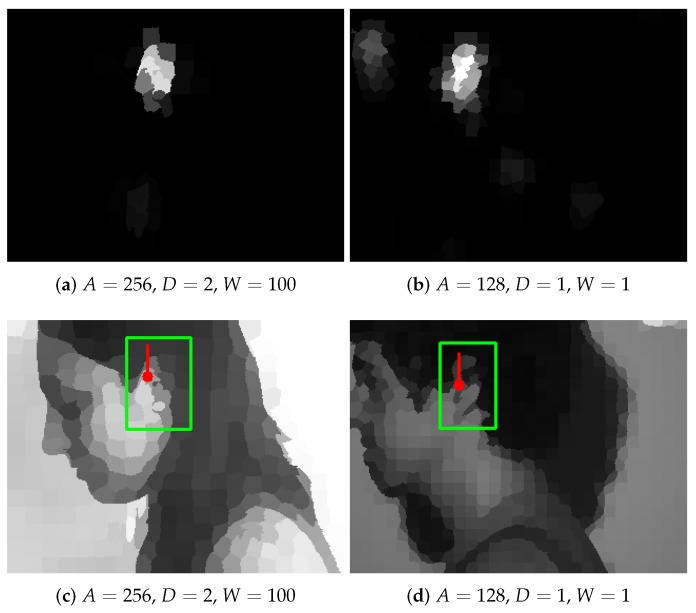
Detection results for images shown in Figure 3: (**a**,**b**) network output (it was scaled back using *W* and cut to [0,255] interval); (**c**,**d**) detection visualization (the green rectangle represents the expected bounding box, and the red dot indicates selected superpixel; the red line shows the orientation of filters in the network, and the vertical line corresponds to basic orientation).

**Figure 10 sensors-19-05510-f010:**
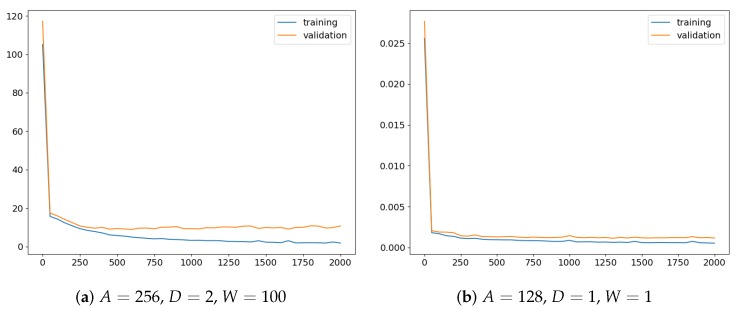
Training characteristics of two models. The plots present errors for training $D_{M}^{TR}$ and validation $D_{M}^{VA}$ sets that were calculated every 50 epochs. On the left, the training run for the best parameter combination is depicted. It can be observed that to select the optimal network, the model generated after 600 epochs should be chosen. On the right, another combination of parameters was used. This time, however, no epoch can be indicated where model overfitting seemed to take place. Probably, the training should be continued further.

**Figure 11 sensors-19-05510-f011:**
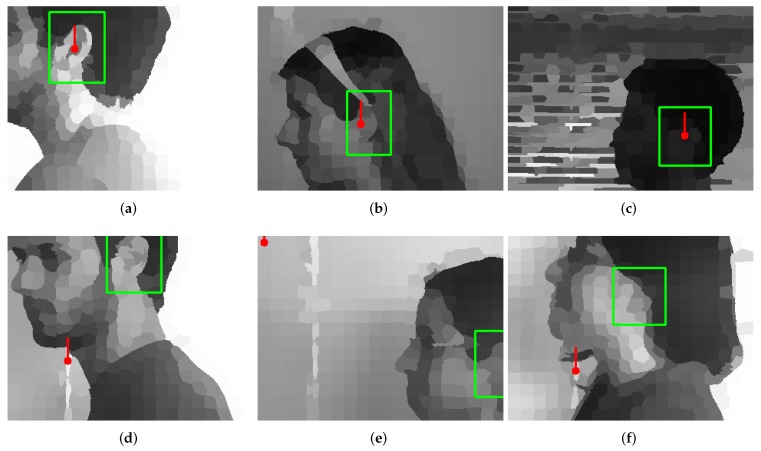
Examples of detections for the best, trained network for images in $D_{M}^{VA}$ and $D_{M}^{TE}$: (**a**–**c**) correct detections; (**d**–**f**) wrong detections. The network is able to respond correctly in different illumination conditions and for different backgrounds. Problems can be observed when ears are not fully visible (image border or occlusion) and when the head orientation is different than expected (wrong annotation). The convention of the result presentation was described in Figure 9’s caption.

**Figure 12 sensors-19-05510-f012:**
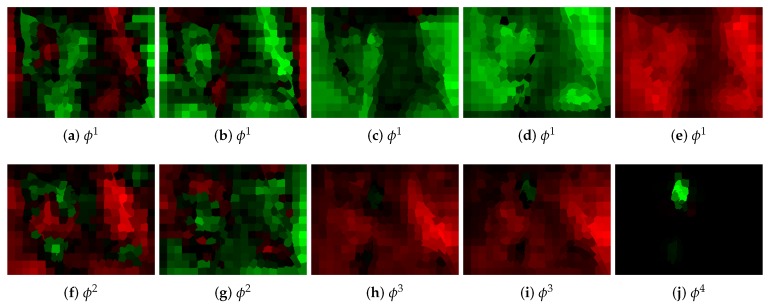
Selected, raw outputs of convolutional layers for the image shown in Figure 3a and the optimal network. In the first two layers ($\phi^{1}$ and $\phi^{2}$), the person outlines can be still observed, so probably some local image characteristics are extracted here. This behavior is typical also for classic CNNs. In the final layer $\phi^{4}$, the output allows finding ear position. For visualization purposes, every output was scaled separately (they cannot be compared). Red color denotes negative and green positive values.

**Figure 13 sensors-19-05510-f013:**
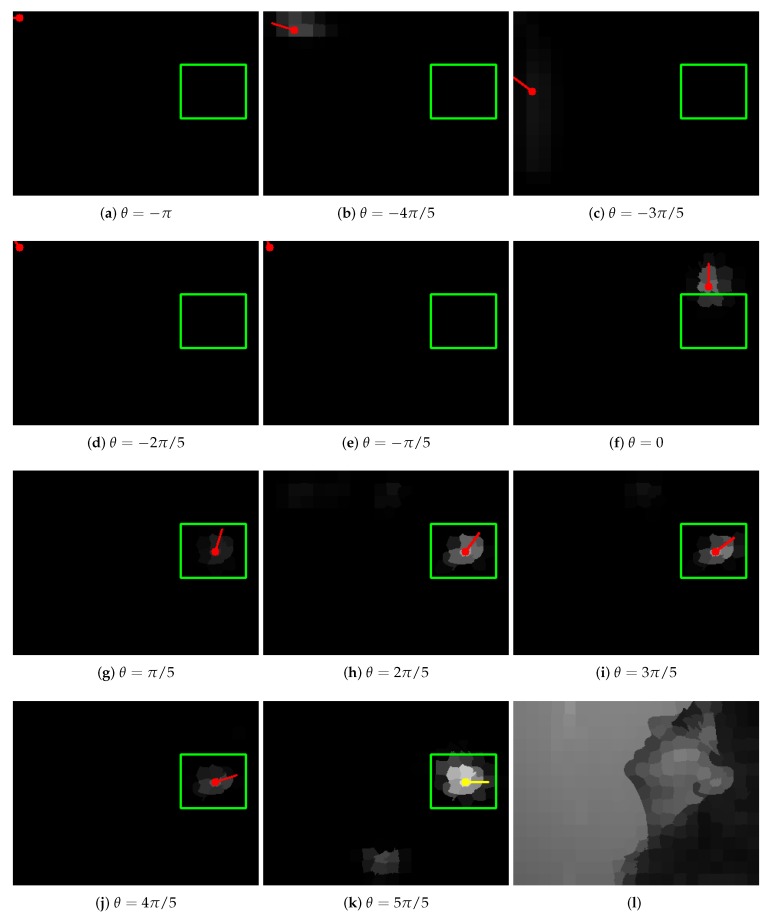
Pictures (**a**–**k**) present the outputs of successive networks $\Phi_{\theta }$. The green rectangle shows the expected ear location. Yellow and red dots with a line indicate superpixels with maximum value (yellow identifies the maximum among all the outputs; the line shows the orientation of filters in the network). The last picture (**l**) shows the input image. Starting from angle θ=π/5, networks are able to detect ear correctly. In (**b**,**k**), unexpected artifacts can be noticed. Picture (**f**) demonstrates an alternative detection region, which, when observed locally, can be indeed mistakenly recognized as an ear.

**Figure 14 sensors-19-05510-f014:**
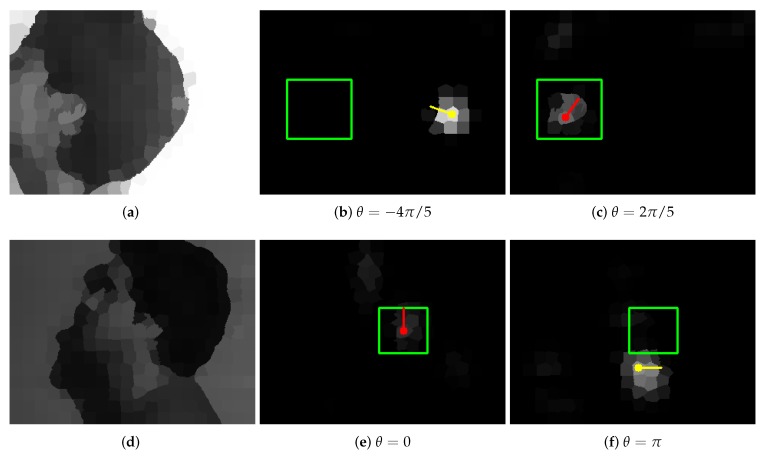
Selected outputs of networks $\Phi_{\theta }$ for images where the maximum node value does not allow detecting ear correctly: (**a**,**d**) input image; (**c**,**e**) output with correct detection; (**b**,**f**) output with the node having the maximum value, wrong detection. The unexpected local maxima can be observed in areas of uniform color. The convention of the result presentation was described in Figure 13 caption.

**Table 1 sensors-19-05510-t001:** MSE errors of network $\Phi_{0}$ (trained using $D_{0}^{TR}$ to detect horizontal edges) and its rotated versions $\Phi_{\theta }$ calculated for datasets $D_{\theta }$ with different expected orientation of edges. As was expected, when the edge rotation corresponds to network (filter) rotation, the MSE errors are significantly smaller than errors obtained for the original network $\Phi_{0}$. The presented errors in fact involve only a small number of superpixels as both in network outputs and in expected graphs, most of the values are equal to 0.

θ=		0	3π/4	−π/2	
$\Phi_{0}$	$D_{\theta }^{TR}$	4	32.7	32.9	$\times \;10^{-4}$
	$D_{\theta }^{VA}$	4.1	33	31.7	
	$D_{\theta }^{TE}$	9.3	36	30.5	
$\Phi_{\theta }$	$D_{\theta }^{TR}$	4	6.1	7.2	$\times \;10^{-4}$
	$D_{\theta }^{VA}$	4.1	5.2	7.8	
	$D_{\theta }^{TE}$	9.3	11.4	11.4	

**Table 2 sensors-19-05510-t002:** The ear detection accuracy of the networks trained using the $D_{M}^{TR}$ set for different combinations of parameters *A*, *D*, and *W*. Additionally, the last column contains the cardinality of every considered set. It is worth noticing that networks trained using only samples for basic head orientation (pose M) can successfully detect ear not only for other people, but also for different orientations (poses U and D). Naturally, in the latter case, detection accuracy was significantly smaller.

A=	256				128				#D
D=	1		2		1		2		
W=	1	100	1	100	1	100	1	100	
$D_{M}^{TR}$	99.52%	97.83%	100%	99.76%	98.07%	98.07%	100%	100%	415
$D_{M}^{VA}$	91.85%	97.04%	94.7%	97.04%	92.59%	93.33%	93.33%	95.56%	135
$D_{M}^{TE}$	92.86%	96.1%	90.91%	98.05%	94.16%	93.51%	99.35%	96.75%	154
$D_{U,D}^{TR}$	73.73%	75.36%	70.88%	80.45%	74.95%	77.19%	80.45%	79.23%	491
$D_{U,D}^{VA}$	79.19%	81.88%	73.15%	87.92%	78.52%	77.18%	85.23%	81.88%	149
$D_{U,D}^{TE}$	74.73%	83.33%	77.42%	81.72%	77.42%	74.73%	82.26%	81.18%	186

**Table 3 sensors-19-05510-t003:** Comparison of the detection accuracy for different subsets of the UBEAR dataset and different detection models. The reference solution is network $\Phi_{0}$, which was the best network trained using only only samples from $D_{M}^{TR}$ (basic head orientation). The second column represents the approach where the maximum of networks $\Phi_{\theta }$ outputs indicates the ear position. In the third column, detection was considered successful if at least one $\Phi_{\theta }$ output showed correct ear position. It presents the maximum possible accuracy if the correct θ can be found. The fifth and sixth column contain results of models trained using images with all head poses gathered in set $D_{ALL}^{TR}$. Network $\Phi_{ALL}$ has the same architecture as $\Phi_{0}$ (4 layers), whereas $\Phi_{ALL}^{+}$ has an additional layer at the beginning (5 layers). The last column contains the cardinality of every considered set.

	max $\Phi_{\theta }$	any $\Phi_{\theta }$	$\Phi_{0}$	$\Phi_{\mathrm{ALL}}$	$\Phi_{\mathrm{ALL}}^{+}$	#D
$D_{M}^{TR}$	94.22%	100%	99.76%	99.28%	100%	415
$D_{M}^{VA}$	83.7%	100%	97.04%	94.81%	100%	135
$D_{M}^{TE}$	84.42%	99.35%	98.05%	95.45%	95.45%	154
$D_{U,D}^{TR}$	62.93%	96.54%	80.45%	99.39%	99.8%	491
$D_{U,D}^{VA}$	69.8%	98.66%	87.92%	97.99%	97.32%	149
$D_{U,D}^{TE}$	71.51%	98.82%	81.72%	94.62%	97.85%	186
$D_{ALL}^{TR}$	67.67%	93.09%	83.25%	98.75%	99.56%	1361
$D_{ALL}^{VA}$	68.79%	97.87%	88.89%	92.91%	96.45%	423
$D_{ALL}^{TE}$	67.45%	93.37%	81.87%	94.15%	94.15%	513

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
