# Peer review of "Ear Detection Using Convolutional Neural Network on Graphs with Filter Rotation"

_sensors, 2019, doi:10.3390/s19245510_

Round 1

Reviewer 1 Report

The authors presented a method for ear detection with traditional segmentation algorithm and learning-based approach. The manuscript is well-organized. The figures are with good quality. The proposed method is useful for researchers in the field of human recognition.

If I need to make some suggestions, I think Chapter 1 and 2 can be combined together, since related method overview should be part of introduction.

Author Response

Dear Sir or Madam,

At the beginning we would like to express our gratitude for your valuable comments.

Following your suggestion sections 1 and 2 were combined together. Subsections 2.1, 2.2 and 2.3 were moved to subsections 1.1, 1.2 and 1.4, respectively. In section 1.3 the part of introduction devoted to geometric deep learning was placed.

All other text modifications were highlighted in blue. They are either a consequence of the reviewers’ comments or corrections of typos.

We have also added three additional illustrations. These are figures 1, 2 and 4. They should allow the reader to understand better both general algorithmic flow and operation of single convolutional layer.

Yours faithfully,

Authors

Reviewer 2 Report

The following are the critical remarks:

1 - In section 3, before 3.1, the general algorithmic flow must be presented, perhaps, bu using a flow chart or a diagram. In the present form, section 3 does not seem to combine the parts of the approach together, and it seems like description of separate approaches.

2 - subsection 3.2 seems incomplete or does not have a conclusive sentence. The purpose of section 3.3. described the experiment organization rather than "verification" pf the used architecture of the CNN.

4 - Some phrases must be corrected such as in line 82-83: "As it was mentioned ????, in this section, first related works ???? concerning ear detection and rotation invariance/equivariance are presented."

5 - Figures 3 and 7 are hard to read due to small fonts.

6 - The Summary is not well written. Some phrases must be clarified, such as the last few sentences:

"Reduced image content representation should be very helpful here since there is no need to operate on huge set of pixels. That potential of explainability is an additional advantage of presented technique."

It is not clear what was the intention of the authors here regarding the "context" of the picture and human perception versus machine's approach.

Author Response

Dear Sir or Madam,

At the beginning we would like to express our gratitude for your valuable comments.

We have taken into account all your remarks. Detailed description of changes is presented at the end of this letter. All text modifications were highlighted in blue. They are either a consequence of the reviewers’ comments or corrections of typos.

Following suggestion of one of the reviewers sections 1 and 2 were combined together. Subsections 2.1, 2.2 and 2.3 were moved to subsections 1.1, 1.2 and 1.4, respectively. In section 1.3 the part of introduction devoted to geometric deep learning was placed.

We have also added three additional illustrations. These are figures 1, 2 and 4. They should allow the reader to understand better both general algorithmic flow and operation of single convolutional layer.

Detailed response:

1 - In section 3, before 3.1, the general algorithmic flow must be presented, perhaps, bu using a flow chart or a diagram. In the present form, section 3 does not seem to combine the parts of the approach together, and it seems like description of separate approaches.

After paper reorganization section 3 became a section 2. The introductory part of this section was extended and Figure 1 was added to present general algorithmic flow (its caption explains all the processing steps). Moreover, the end of subsection 2.2 and the beginning of subsection 2.3 were modified to explain their relationship. New figures 2 and 4 should also allow to better understand the presented ideas.

2 - subsection 3.2 seems incomplete or does not have a conclusive sentence. The purpose of section 3.3. described the experiment organization rather than "verification" pf the used architecture of the CNN.

After combining of sections 1 and 2 subsections 3.2 and 3.3. became subsections 2.2 and 2.3 respectively. We have significantly extended the last paragraph of subsection 2.2 illustrating it with figure 2. After this modification and after changing the first sentence of section 2.3, we believe, its purpose is more clear. It simply verifies the hypothesis about the properties of considered network. This hypothesis can be formulated as follows: rotation of the trained filters causes that network can identify rotated structures.

4 - Some phrases must be corrected such as in line 82-83: "As it was mentioned ????, in this section, first related works ???? concerning ear detection and rotation invariance/equivariance are presented."

Indeed this phrase was not clear. Due to reorganization of sections 1 and 2 it became unnecessary (it was removed) as it was a part of introductory paragraph of old section 2.

5 - Figures 3 and 7 are hard to read due to small fonts.

We have enlarged those fonts.

6 - The Summary is not well written. Some phrases must be clarified, such as the last few sentences: "Reduced image content representation should be very helpful here since there is no need to operate on huge set of pixels. That potential of explainability is an additional advantage of presented technique." It is not clear what was the intention of the authors here regarding the "context" of the picture and human perception versus machine's approach.

To support our claims, in particular there, where the summary of results is presented, we have added some references to figures and tables. Moreover, to explain the sentence cited above, which in fact is not clear, we have significantly extended the last paragraph of this section.

Yours faithfully,

Authors

Reviewer 3 Report

I accept the paper in its present form. It is well written. Its contribution lies in:

application of Geometric deep learning (GDL) to semantic segmentation of images, introduction of trained filters rotations; the filters are defined by Gaussian mixture model - this avoids interpolation problems, when filters are rotated, using superpixel representation, using the methods for ear detection (important and attractive in biometrics), showing their usefulness on the representative dataset (UBEAR), and proving, that training of the proposed model with limited number of samples and rotation of trained filters allows to detect the rotated structures.

Minor remarks: persons –people?, line 98 sstep, 127 uses, 247 andle, 269 introduces, 295 DVA.

Author Response

Dear Sir or Madam,

At the beginning we would like to express our gratitude for your valuable comments.

We have taken into account all your remarks. These changes and all other text modifications were highlighted in blue. They are either a consequence of the reviewers’ comments or corrections of typos.

Following suggestion of one of the reviewers sections 1 and 2 were combined together. Subsections 2.1, 2.2 and 2.3 were moved to subsections 1.1, 1.2 and 1.4, respectively. In section 1.3 the part of introduction devoted to geometric deep learning was placed.

We have also added three additional illustrations. These are figures 1, 2 and 4. They should allow the reader to understand better both general algorithmic flow and operation of single convolutional layer.

Yours faithfully,

Authors